# Ensemble Capsule Network with an Attention Mechanism for the Fault Diagnosis of Bearings from Imbalanced Data Samples

**DOI:** 10.3390/s22155543

**Published:** 2022-07-25

**Authors:** Zengbing Xu, Carman Ka Man Lee, Yaqiong Lv, Jeffery Chan

**Affiliations:** 1Centre for Advances in Reliability and Safety, Hong Kong; zengbing.xu@cairs.hk (Z.X.); jeffchchan@gmail.com (J.C.); 2School of Machinery and Automation, Wuhan University of Science and Technology, Wuhan 430081, China; 3Department of Industrial and Systems Engineering, The Hong Kong Polytechnic University, Hong Kong; 4School of Transportation and Logistics Engineering, Wuhan University of Technology, Wuhan 430062, China; lvp@sdju.edu.cn

**Keywords:** ensemble capsule network, imbalanced dataset, fault diagnosis

## Abstract

In order to solve the problem of imbalanced and noisy data samples for the fault diagnosis of rolling bearings, a novel ensemble capsule network (Capsnet) with a convolutional block attention module (CBAM) that is based on a weighted majority voting method is proposed in this study. Firstly, the complete ensemble empirical mode decomposition with adaptive noise (CEEMDAN) method was used to decompose the raw vibration signal into different IMF signals, which are noise reduction signals. Secondly, the IMF signals were input into the Capsnet with CBAM in order to diagnose the fault category preliminarily. Finally, the weighted majority voting method was utilized so as to fuse all of the preliminary diagnosis results in order to obtain the final diagnostic decision. In order to verify the effectiveness of the proposed ensemble of Capsnet with CBAM, this method was applied to the fault diagnosis of rolling bearings with imbalanced and different SNR data samples. The diagnostic results show that the proposed diagnostic method can achieve higher levels of accuracy than other methods, such as single CNN, single Capsnet, ensemble CNN and an ensemble capsule network without CBAM and that it has stronger immunity to noise than an ensemble capsule network without CBAM.

## 1. Introduction

Bearings work as key components of rotary machinery, as such their states directly affect a machine’s performance. When they work for a long time under variable working conditions, they can suffer various faults [1,2,3]. Fault diagnosis is an effective measure that is used to maintain the normal operation of mechanical equipment. However, in real industrial production, the amount of normal data within the sample is greater, the amount of fault data within the sample is lesser and the data sample is contaminated by noise [4]. It is very important to be able to diagnose these bearing faults under the conditions of imbalanced data samples and strong noise [5].

Deep learning has been used to reform intelligent fault diagnosis due to its end-to-end diagnostic ability, which can employ a hierarchical network in order to extract the fault features from the raw data, layer by layer, automatically, rather than requiring artificial feature extraction. In this process, the output layer is replaced by an artificial neural network (ANN)-based classifier due to its high capability in multiclass classification in the diagnosis of the health states of bearings [5]. Due to these merits, many deep learning models have been suggested for application to the fault diagnosis of mechanical equipment, such as the denoising autoencoder (DAE), deep belief network (DBN), convolutional neural network (CNN) and so on [5,6,7,8,9].

It is well known that the deep learning-based diagnostic method needs large-scale and balanced training data samples [5]. Imbalanced data samples can affect the diagnostic performance of deep learning models [10]. In order to solve this problem, some data augmentation methods have been proposed to increase the prevalence of the fault-related data in the samples. For example, a weighted minority oversampling method that has been proposed used resampling measures to generate the faulty data samples in order to balance the data distribution [8]. Afterwards, a deep learning-based data augmentation strategy was proposed to expand the limited fault samples [11], examples of this include the generative adversarial network (GAN) and variants of this that are comprised of a generator (G) network and a discriminator (D) network which were developed to generate data samples, with the same data distribution as the original data, from random noise [12,13,14,15,16,17,18,19,20]. Nevertheless, the disadvantage of these data augmentation methods is that some unnecessary or incorrect samples can be generated and the sample’s diversity cannot be augmented, which reduces the quality of the generated samples and has a very limited level of improvement for the performance of the diagnostic model [13,18].

Ensemble learning, which has been verified for its use in overcoming the limitations of data sample imbalances, achieves high diagnostic accuracy and generalization because of its complementary diagnostic behavior among different diagnostic models without generating additional fault data samples. The greater the difference between versions of a single diagnostic model is, the better the performance of the ensemble diagnosis models is [21]. So, when multiple diagnostic models that are combined with the different signals are applied in the field of fault diagnosis, the resulting ensemble diagnostic models can not only reduce the effect of imbalanced data samples and noise, but also achieve a relatively higher level of diagnostic accuracy and generalization [22].

Currently, CNN is widely applied to the field of pattern recognition and fault diagnosis because of its strong spatiotemporal feature extraction capability. Capsule network (Capsnet) can be recognized as an improved version of CNN that can be used to classify categories with small data samples, as it can not only extract the feature vector but also depict the spatial relative-positional relationships of objects [17,18,19,20]. Due to these abilities, it is theoretically possible to perform the diagnostic analysis of faults with a small number of samples. Owing to these characteristics, Capsnet and its variants have been developed for the application to the fault diagnosis of bearings [23,24,25,26,27]. For example, the Bi-LSTM and Capsule network with CNN are used effectively to diagnose bearing faults with insufficient fault data samples [24]. Furthermore, a novel capsule network with an inception block and a regression branch has been proposed for the diagnosis of bearing faults with high accuracy and good generalization [27]. Although Capsnet can effectively diagnose faults to a certain extent, Capsnet is prone to perturbation by the data sample’s imbalance and strong noises. At present, some ensemble Capsnet models have been proposed that may be used to diagnose bearing faults. For example, a Capsnet model that is based on sensor fusion can effectively diagnose bearing faults [28]. A novel Capsnet model that is based on wide convolution and multi-scale convolution (WMSCCN) using AdaBN has been used to diagnose bearing faults with varying sizes of training samples, having strong robustness and high accuracy [29]. These ensemble Capsnet models mainly fuse multi-sensor signals or different scale features that have been extracted by different convolution kernel sizes in order to diagnose the faults, but Capsnet can only capture the global information from the original vibration signal and it ignores noise effects and the fault information that is concealed in the local signal, which can result in low diagnostic accuracy. 

In order to capture more of the fault-related information that is concealed in the raw vibration signal and to reduce the impact of noise, different kinds of time frequency analysis methods, such as the short-time Fourier transform (STFT), empirical mode decomposition (EMD) and wavelet packet transform (WPT), have been proposed to decompose the vibration signal into its different scale components so that a deep learning model can capture the local fault information from the different scale components for fault diagnosis [30,31]. It is well known that the intrinsic model function (IMF) signal of EMD can not only describe the physics mechanism of a dynamic signal to some extent, but it can also be identified as creating a noise-reduced signal as the whole decomposition process of EMD reflects a multi-scale filtering process. Considering the mode mixing of the different IMF signals of traditional EMD and ensemble EMD (EEMD), a complete ensemble EMD with adaptive noise (CEEMDAN) has been adopted in order to decompose the raw signal into different IMF signals which depict the vibration characteristics from the high frequency to the low frequency more accurately [32,33,34,35,36]. However, in the diagnostic process, the different IMF signals that are fed into Capsnet have different contributions to the diagnostic results. The allocations of these signals’ weights can affect the diagnostic accuracy.

In addition, in order to further improve the capability of fault-sensitive feature mining and the diagnostic accuracy of Capsnet, a convolutional block attention module (CBAM) [37] has also been introduced into Capsnet, which can capture some sensitive fault-related information from different channels and spatial locations. According to the above analyses and discussion, a novel ensemble Capsnet model, which integrates a modified EEMD and CBAM, has been suggested to diagnose rolling bearing faults. The innovation and contribution of the proposed diagnostic method are summarized as follows:(1)The ensemble Capsnet that is based on a weighted majority voting method has been suggested to diagnose the imbalanced bearing fault data samples with high accuracy and strong robustness, which can not only consider the different degrees of importance of the IMF components to the diagnosis results but can also fuse all of the preliminary diagnostic results that were obtained by all of the single Capsnet models combined with the individual IMF.(2)The single Capsnet model can extract hidden feature parameters from the different IMF signals which were denoised and decomposed by CEEMDAN, so as to capture more fault information from the different scales in order to improve the diagnostic accuracy.(3)The CBAM can select fault-sensitive features that are extracted by the Capsnet in order to further improve the diagnostic accuracy.

The remainder of this paper is organized as follows. The proposed ensemble capsule network model, which synthesizes the modified EMD method and CBAM, is described in Section 2. Section 3 shows the fault diagnostic analysis of bearings. The conclusions are drawn in Section 4.

## 2. The Proposed Ensemble Capsnet with CBAM

In order to make use of the merits of Capsnet and ensemble learning, an ensemble capsule network with CBAM, based on the weighted majority voting method (WMAM), is proposed for the diagnosis of bearing faults with imbalanced data samples. Figure 1 shows the schematic diagnosis flowchart. The raw vibration signal was gradually truncated through a sliding time window and segmented into N data samples, then the modified EMD was utilized in order to decompose these data samples into multiple denoised IMF signals. Afterwards, the IMF signals were input into Capsnet with CBAM in order to extract sensitive features and to diagnose the fault preliminarily. Finally, the final diagnostic decision was made by the application of the weighted majority voting method.

### 2.1. CEEMDAN

The CEEMDAN is an improved version of EMD and ensemble EMD (EEMD), which can decompose a vibration signal into multiple eigenmode functions in order to reduce its fluctuation and to denoise the signal [26]. Presently, this method has been widely utilized to process nonlinear and nonstationary signals due to its ability to overcome the mode aliasing problem of EMD and the difficulty of determining the amplitude of Gaussian white noise in the EEMD method. In order to capture and understand more fault-related information from the different scales, CEEMDAN can be adopted in order to decompose the raw vibration signal into different intrinsic mode functions (IMFs). The detailed algorithm of CEEMDAN can be described as follows.

Step 1: Assuming x(n) is the raw input signal, A_0_ is an amplitude coefficient of the white noise, $v_{i\;}\left(n\right)$ is the Gaussian white noise sequence, the i-th signal sequence is $x^{i}\left(n\right)=x\left(n\right)+A_{0}v_{i}\left(n\right)$ (i = 1, 2, 3⋯, I, I is the counts of added white noise), $E_{K}\left(\cdot \right)$ is the modal component of the k^th^ order produced by EMD method and the k-th modal component that was decomposed by the use of the CEEMDAN method is denoted as ${\mathrm{CIMF}}_{k}\left(\cdot \right)$.

Step 2: White noise $A_{0}v_{i}\left(n\right)$ is added to the original input signal x(n) and the first modal component can be obtained by the use of EMD decomposition, as follows:(1)$${\mathrm{CIMF}}_{1}\left(n\right)=\frac{1}{I}\sum_{i=1}^{I}{\mathrm{IMF}}_{1}^{i}\left(n\right)$$
the corresponding residual signal $r_{1}\left(n\right)=x\left(n\right)-{\mathrm{CIMF}}_{1}\left(n\right)$ of the first order (k = 1) is also obtained.

Step 3: The first-order residual signal $r_{1}\left(n\right)$ is added to the IMF component $A_{1}E_{1}\left(v^{i}\left(n\right)\right)$, where $E_{1}\left(v^{i}\left(n\right)\right)$ is the first-order modal component of the white noise $E_{1}\left(v^{i}\left(n\right)\right)$ which is decomposed by the use of the EMD method. Thus, the newly formed signal $r_{1}\left(n\right)+A_{1}E_{1}\left(v^{i}\left(n\right)\right)$ can be decomposed in order to obtain the second modal component as follows:(2)$${\mathrm{CIMF}}_{2}\left(n\right)=\frac{1}{I}\sum_{i=1}^{I}E_{1}(r_{1}\left(n\right)+{\;A}_{1}E_{1}\left(v^{i}\left(n\right)\right)\;$$

Step 4: The k-th residual signal $r_{k}\left(n\right)=r_{k-1}\left(n\right)-{\mathrm{CIMF}}_{k}\left(n\right)$ of the other orders (k = 2, 3⋯, K) can be similarly obtained according to step (3). Thus, the (k + 1)-th modal component is calculated as follows:(3)$${\mathrm{CIMF}}_{k+1}\left(n\right)=\frac{1}{I}\sum_{i=1}^{I}E_{1}(r_{k}\left(n\right)+{\;A}_{k}E_{k}\left(v^{i}\left(n\right)\right)$$

Step 5: Repeat step 4 until the decomposition conditions of the EMD method are not met by the remaining components. Finally, the raw vibration signal can be decomposed into different IMFs, as follows:(4)$$x\left(n\right)=\sum_{i=1}^{k}{\mathrm{CIMF}}_{i}\left(n\right)+R\left(n\right)$$
where R(n) is the residual signal.

### 2.2. Capsnet with CBAM

Capsnet can not only extract the features from the IMF signal but it can also conserve the spatiotemporal relationship between different features. CBAM can utilize the attention mechanism in order to select the sensitive features so as to form a feature map wherein the features are more representative. In order to enhance the accuracy of the fault diagnosis, Capsnet combined with CBAM was used to extract the sensitive features so as to diagnose a bearing fault.

#### 2.2.1. Capsule Network

Capsnet is a modified version of CNN that was introduced by Sabour et al. [19]. In order to reduce information loss and improve its feature extraction ability, the CNN’s scalar-in and scalar-out mechanism was substituted with a vector-in and vector-out mechanism from Capsule. The basic framework of the original Capsnet is mainly comprised of a convolutional operation and dynamic routing agreement and it consists of three typical layers: the convolutional layer, the primary capsule layer and the digital capsule layer, as presented in Figure 2. 

The convolutional operation uses convolution kernels to convert the raw input data into the local feature maps that are utilized as the inputs for the primary capsule layer by the nonlinear activation function, which can be written as follows:(5)$$H_{q}=f\left(\sum_{p}x_{p}*W_{pq}+b_{q}\right)$$
where $x_{p}$ is the input feature maps, $H_{q}$ is output flows, $W_{pq}$ and $b_{q}$ are the weights and biases, respectively, $f\left(\cdot \right)$ is a nonlinear activation function ReLU and the notation * denotes convolution calculation. 

After the convolutional operation, the relationship between the primary capsule layers and digital capsule layers can be built by the utilization of a dynamic routing algorithm that is more effective for learning discriminative representations than the pooling operation in CNN due to its ability to identify the position of one feature relative to another. Therefore, the length of the activity vector of each digital capsule depicts the presence probability of an instance for each category, which is equivalent to the categories to be classified. The calculation process of the dynamic routing algorithm is shown in the Figure 3. Assuming $U^{l}=\left[u_{1}^{l},u_{2}^{l},\ldots ,u_{H}^{l}\right]$ are all neurons in the primary capsule layer and that the total input to the digit capsule $S_{j}$ is the weighted sum of all of the middle prediction vectors $m_{j|i}$ from the capsules in the primary capsule layer, then $m_{j|i}$ can be calculated by multiplying the neuron $u_{i}^{l}\in U^{l}$ in the primary capsule layer with a transformation matrix $W_{ij}$, which is described as: (6)$$m_{j|i}=W_{ij}u_{i}^{l}$$
and the input vector $S_{j}$ can be obtained by the application of the following formula:(7)$$s_{j}=\sum_{i}c_{ij}m_{j|i}$$

The output vector $v_{j}$, which is the output vector of the higher-level capsule j, can be calculated by the nonlinear mapping of $s_{j}$, as follows:(8)$$v_{j}=Squashing\left(s_{j}\right)$$
(9)$$Squashing\left(s\right)=\frac{{||S||}^{2}}{1+{||S||}^{2}}\times \frac{S}{||S||}=\frac{S||S||}{1+{||S||}^{2}}$$
where the subscript *j* represents the *j*th output neuron and $c_{ij}$ is a coupling coefficient that is amended by the iterative process of a dynamic routing agreement algorithm during training and can be updated using the following function
(10)$$c_{ij}=softmax\left(b_{ij}\right)=\exp\left(b_{ij}\right)/\;\sum \exp\left(b_{i.}\right)$$
where the deviation $b_{ij}$ denotes the log prior probability of the coupling coefficient that capsule *i* couples with capsule *j*, which can be updated by the following equation
(11)$$b_{ij}=b_{ij}^{\prime }+u_{ij}$$
where $b_{ij}^{\prime }\;$is the previous value and the “agreement” $u_{ij}$ is defined as follows:(12)$$u_{ij}=\langle v_{j},m_{j|i}\rangle$$

Thus, with the iterations from (6) to (12), all of the relevant parameters and the agreement-based dynamic routing algorithm of Capsnet are determined. The more detailed algorithm of Capsnet can be seen in [19].

#### 2.2.2. Convolutional Block Attention Module (CBAM)

In order to select the sensitive features from the feature maps that are extracted by the convolutional operation, the attention mechanism was used to increase the representational power of the important features and to suppresses that of the unnecessary ones. In order to gain the benefit from extracting discriminative features, a CBAM was adopted to focus on channel information and spatial information at the same time, as first proposed by Woo et al. [37]. The structure is shown in Figure 4, it consists of a channel attention process and a spatial attention process. The channel attention map Mc was obtained in the channel attention mechanism through the selection of the channel and the spatial attention map Ms was obtained in the spatial attention mechanism by the selection of the sensitive features of the channel. When the input feature F passes through these two attention modules in order to obtain the refined feature F”, the selection process of the features can be represented by the following two equations.
(13)$$F’=M_{c}\left(F\right)⊗F$$
(14)$$F”=M_{s}\left(F\prime \right)⊗F\prime$$
where $F\in R^{C\times H\times B}$ is the input features map of the CBAM module with the channel number C, the height H and the width W. F′ denotes the feature map multiplying the channel attention map and F″ is the result of the spatial attention map multiplying F′, which denotes the output of CBAM module. $M_{c}\in R^{C\times 1\times 1}$ denotes the attention weight in the channel dimension. $M_{s}\in R^{1\times H\times B}$ denotes the attention weight in the spatial dimension. The symbol ⊗ represents element-wise multiplication. The detailed algorithm can be seen in reference [10]. 

#### 2.2.3. Diagnosis Based on the Capsule Network with CBAM

Based on the characteristics of Capsnet and CBAM, the convolutional operation in Capsnet was used to extract the feature parameters and the CBAM was utilized to select the sensitive feature parameters that were input into the capsule network so as to improve the diagnostic performance of Capsnet. The overall structure of the Capsnet with CBAM model is shown in Figure 5. The raw data segment, which contains 1024 data points, was decomposed into different scale IMF signals. The 1-D IMF signal was reshaped into the 2-D grey maps of 32×32 size [29], then the Conv layers (of 3×3 kernel size) and the average pooling layers (of 2 × 2 size) were used to extract the feature parameters and the CBAM was used to encode where to emphasize or suppress the feature parameters. In addition, in order to avoid gradient vanishing and improve the nonlinear ability of model, the Relu function was selected as the activation function in all of the convolution layers. After that, these selective feature parameters were reshaped and input into Capsnet in order to diagnose the bearing fault category preliminarily. 

In the training process of Capsnet, so as to obtain the optimal weight parameters, the margin loss function was defined as follows:(15)$$L_{k}=T_{k}\max{\left(0,m^{+}-||a^{k}||\right)}^{2}+\lambda \left(1-T_{k}\right)\max{\left(0,a^{k}-m^{-}\right)}^{2}$$
where k denotes the fault class, $T_{k}$ denotes the indicator function and, if class k is present, $T_{k}=1$, otherwise 0. The value $m^{+}$ denotes the upper bound that is be used to punish false positives, $m^{-}$ denotes the lower bound that is used to punish false negatives and λ denotes the coefficient. The values of these corresponding parameters $m^{+}$*,*$\;m^{-}$ and λ were set as 0.9, 0.1 and 0.5, respectively. The value$\;a^{k}$ is the probability value of the fault class k, which cannot be less than 0.9 if the fault class k is present. Conversely, if the fault class k is not present, then $||a^{k}||$ cannot be greater than 0.1.

### 2.3. The Weighted Majority Voting Method (WMVM)

In order to improve the diagnostic accuracy and robustness of Capsnet with ensemble learning using the different IMF signals, CBAM was integrated with multiple classifiers so as to develop an ensemble Capsnet for use in diagnosing the faults in parallel. Considering that these IMF signals have different contributions to the diagnostic results, all of the preliminary diagnosis results of the differently scaled signals were fused in order to obtain the final class label by the weighted majority voting method (WMVM) in the decision making level, so as to have high accuracy in the final diagnostic result [30]. Equation (16) formulates the final diagnosis operations, as follows.
(16)$$H\left(x\right)=C_{armax_{j}}\overset{N}{\underset{n=1}{\sum }}w_{n}h_{n}^{j}\left(x\right)\;$$
where, for each data sample x, the final prediction class label $H\left(x\right)$ can be calculated by the application of the function $C_{armax_{j}}\left(\cdot \right)$ in order to find out which prediction probability has the maximum vote value. The value $h_{n}^{j}\left(x\right)$ is the prediction probability of the N sub-classifier, the weight for majority voting of each $h_{n}^{j}\left(x\right)$ is $w_{n}$ (which directly affect the final diagnostic results) and $w_{n}$ can be calculated by the use of the equation $w_{n}=c_{n}/\sum_{n=1}^{N}c_{n}$ in which $c_{n}$ is the validation accuracy of the *n*^th^ sub-classifier that diagnoses the validation dataset. In addition, *N* is set as 7, representing 7 sub-classifiers, and j=1,2,3 is the fault class label in this paper.

### 2.4. Fault Diagnosis Flowchart Based on the Ensemble Capsnet

Figure 6 presents the fault diagnosis flowchart that is based on the proposed ensemble Capsnet model. After obtaining the bearing vibration signal, the raw vibration signal was segmented into different data samples by the use of a sliding time window and then these data segments were divided into three datasets which were the training dataset, validation dataset and test dataset. All of these were decomposed into different IMF signals by the CEEMDAN method. The IMF signals in the training dataset were used to train the Capsnet models and the validation dataset was input into the trained Capsnet models in order to build the ensemble Capsnet which was able to obtain the final diagnostic result by using the weighted majority voting method to fuse the preliminary diagnosis result from each Capsnet. Subsequently, the number of Capsnet models was determined by the number of the IMF signals; the parameters of each Capsnet model were trained by its respective IMF signal.

## 3. Fault Diagnosis of Bearings

In order to verify the validity and reliability of the proposed ensemble Capsnet diagnosis method, the vibration signals were obtained from a dataset of rolling element bearings [38]. The experimental test rig, which basically consisted of a high-speed spindle. The body of the spindle was fixed to the single and extremely rigid support which rested on a massive steel base plate. The same plate carried a couple of supports for the outer rings of two identical roller bearings. The inner rings of these bearings were connected to a very short and thick hollow shaft. The outer ring of bearing B2 was linked to a precision sledge, the motion of which was orthogonal to the shaft. When the sledge was pulled through the rotation of a nut, two parallel springs were compressed and produced the required load, which was measured by the load cell. In the laboratory test bed, the same radial force that is generally exerted by the spur gear was replaced by a load that was applied by a third and larger roller bearing. Two tri-axial accelerometers were mounted at position A1. The vibration data were acquired with a 51.2 K/s sample rate. 

### 3.1. Acquisition of Vibration Data

With the use of the experimental setup that is shown in Figure 7, the vibration signals were collected for diagnosis analysis. A single point defect was introduced into the inner race and roller in order to simulate the different fault categories of bearings. The defect’s diameter was 0.45 mm. Each bearing was tested under variable loads (0, 1012, 1006, 1407 and 992 N) and speeds (6000 and 12,000 rpm). The y-axial vibration signal was utilized for the diagnostic analysis of the bearings’ faults. The data statistics are described in detail in Table 1. Five different datasets with different degrees of imbalance and different sizes of training and validation samples are listed and are referred to as datasets A, B, C, D and E. The total number of training samples and validation samples in those 5 datasets were all the same; they were 220, 320, 420, 280 and 480, respectively. The total number of testing samples in these datasets was 1200. Additionally, the raw vibration signal was segmented into different data samples by the application of a sliding time window and the adjacent data samples had no overlapping region. Each data sample had 1024 sample points.

### 3.2. Diagnostic Analysis

Figure 7 shows the original vibration signals of three bearing fault classes. From the figure, it can be seen that it is very difficult to diagnose the fault classes of bearing because of the small amount of difference from the original waveform. In order to obtain these different scale IMF signals, so as to depict the more fault-related information of the bearing from different viewpoints, all of the raw data samples were decomposed into different IMF signals that were input into Capsnet with CBAM by the CEEMDAN method. The decomposition result of the inner race fault signal of the bearing is shown in Figure 8, which consists of eight IMF signals and one residual signal that can describe the various dynamic characteristics from the different scales. From the figure it can also be observed that the first seven orders of the IMF signal contain multiple frequency components and more fault information.

In order to validate the effectiveness of the proposed ensemble of Capsnet with CBAM, based on WMVM, the first seven orders of the IMF signal were used to diagnose the faults in the imbalanced dataset under different working conditions. The seventh order IMF signal and the residual signal were not selected due to their containing less fault information. The corresponding grey maps of the first seven orders of the IMF signal are shown in Figure 9. From these images, it can be observed that the images of different IMF signals look totally different from each other, this provides an intuitive way to depict the fault information from different viewpoints, which can be prone to diagnosis faults by the ensemble Capsnets with high accuracy and generalization, because these seven diagnostic models (which are formed by feeding seven orders of the IMF signal into seven single Capsnets) have big differences. 

Table 2 shows the diagnostic accuracy for five different datasets when using the single Capsnet with IMF signal and ensemble Capsnet. The table shows that the diagnostic accuracy for the different IMF signals is different; the first five orders of the IMF signal can be diagnosed effectively and the diagnosis accuracy of the IMF0 signal is 1, but the diagnosis accuracy for the last two orders of the IMF signal is low. The proposed ensemble Capsnet with CBAM (which was trained on five different training datasets with different imbalance degrees) can diagnose the same testing samples accurately, with all achieving a value of 1. This is mainly because the different IMF signals have different degrees of importance to the diagnostic results and the decomposed IMF0 signal, which is equivalently separated from other interference signals, contains the most fault-related information regarding the bearing. The other first four orders of the IMF signal, which can depict the dynamic characteristics of different scales, contain more fault-related information such that their corresponding diagnostic accuracy is high. When all of the differently scaled IMF signals are used to diagnose the faults, the seven different single Capsnet diagnostic models, the hyper-parameters of which are different as each Capsnet model is trained by different orders of the IMF signal, can be obtained. The ensemble learning process that is based on the WMVM can fuse all of the diagnostic results that have been produced by single Capsnets with CBAM on each of the IMF signals in order to further improve the diagnostic accuracy by making use of the complementary information that is provided by each single Capsnet diagnosis model.

### 3.3. Diagnostic Analysis of a Noisy Dataset

This present research had studied the anti-noise ability of the proposed ensemble Capsnet with CBAM on the dataset A, which had been contaminated by white noise of different intensities. Table 3 and Figure 10 show the diagnostic accuracy of the ensemble Capsnet with CBAM and the ensemble Capsnet without CBAM on different SNR data samples. From the figure and table, it can be seen that the ensemble Capsnet with CBAM and ensemble Capsnet without CBAM can both diagnose the different SNR data samples effectively. When the SNR was −10 db, −1 db, 10 db and 20 db, the accuracy of the ensemble Capsnet with CBAM and ensemble Capsnet without CBAM was 1; however, when SNR was −20 db, the accuracy of the ensemble Capsnet without CBAM was 0.817, which is lower than that of the ensemble Capsnet with CBAM (which was 1). These results show that the ensemble Capsnet with CBAM and the ensemble Capsnet without CBAM can diagnose noisy, imbalanced samples with high accuracy and immunity to noise. The results demonstrate that ensemble learning can not only improve the diagnostic accuracy of Capsnet but also enhance its anti-noise ability. However, when the SNR is smaller than −10 db, the diagnostic accuracy of the ensemble Capsnet without CBAM was less than 1. It was therefore shown that CBAM not only effectively selects sensitive features, but that it also further improves the anti-noise ability and diagnostic accuracy of Capsnet.

### 3.4. Comparison with Others Methods

In order to verify the superiority of the ensemble Capsnet with CBAM method, comparisons were carried out between this method’s diagnostic performance and that of five other diagnostic methods: single CNN, single Capsnet with or without CBAM, ensemble CNN and ensemble Capsnet without CBAM, all of which were conducted on different datasets. Raw data samples and their corresponding IMF signals were used as the input for the single CNN, Capsnet and ensemble CNN and the Capsnet model and the structural parameters of CNN are shown in Table 4. Table 5 and Figure 11 show the corresponding diagnostic results of the six different diagnostic models from five different datasets. From the table and figure, it can be seen that the diagnostic accuracy of the single Capsnet with CBAM is higher than that of a single Capsnet without CBAM and CNN. This demonstrates that the CBAM module can effectively select the sensitive features and improve the diagnostic performance of Capsnet. At the same time, Table 5 and Figure 11 demonstrate that the accuracy of a single Capsnet with CBAM is lower than that of an ensemble CNN and ensemble Capsnet without CBAM and ensemble Capsnet with CBAM and that the diagnostic accuracy of an ensemble Capsnet without CBAM and ensemble Capsnet with CBAM on five different imbalanced datasets are the highest, at 1. These results demonstrate that ensemble learning can improve the diagnostic accuracy of Capsnet and CNN effectively and that the proposed ensemble Capsnet with CBAM has superior performance in the diagnosing of an imbalanced dataset.

From Table 2 and Table 5, it can be seen that the diagnostic accuracy of the single Capsnet with CBAM on the raw vibration signal of the 5 datasets is 0.673, 0.73, 0.649, 0.82 and 0.794. These accuracies are all lower than the diagnostic accuracy that is produced by the single Capsnet model with CBAM on the first IMF signal, which is 1 for all of the datasets. This is because the raw vibration signal is contaminated by noise or other low-frequency signals and the fault information that is concealed in the raw vibration signal cannot be revealed thoroughly by Capsnet, but CEEMDAN can decompose the raw vibration signal into differently scaled signals from which more fault information and the non-stationary dynamic characteristics of the bearing can be derived by the Capsnet. Thus, the single Capsnet can capture more fault-related bearing information from the first IMF signal, which can diagnose the fault more accurately.

In order to illustrate the fact that the different IMF signals have different degrees of importance to the diagnostic results, the ensemble CNN, ensemble Capsnet with CBAM and ensemble Capsnet without CBAM all adopt the voting method (VM) to fuse all of the preliminary diagnostic results in their respective IMF signals and the final diagnostic results are shown in Figure 12 and Table 6. From the figure and table, it can be seen that the accuracy of the ensemble CNN, Capsnet without CBAM and Capsnet with CBAM based on WMVM were higher than that of the ensemble CNN, Capsnet without CBAM and Capsnet with CBAM based on WMVM or VM. Therefore, it can be demonstrated that the different IMF signals have different contributions to the diagnostic results and the weighted fusion method that is based on WMVM can further improve the diagnostic accuracy. 

Furthermore, in order to verify the diagnostic generalization of the proposed ensemble Capsnet with CBAM, the ensemble CNN and the ensemble Capsnet without CBAM, five-fold cross validations were implemented in the analysis of dataset E. Figure 13 shows the diagnostic accuracy, in terms of the mean and variance, of these three ensemble diagnostic models. From Figure 12 it can be seen that the diagnostic accuracy mean of the proposed ensemble Capsnet with CBAM is the highest and the variance is the smallest among these three methods, but the diagnostic accuracy mean of the ensemble CNN is the lowest and the accuracy variance of the ensemble Capsnet without CBAM is the biggest. All of these results can be seen to demonstrate that the proposed ensemble Capsnet with CBAM has strong diagnostic robustness and high diagnostic accuracy and that CBAM can further improve the diagnostic performance of an ensemble Capsnet.

## 4. Conclusions

A novel ensemble Capsnet with CBAM that is based on the weighted majority voting method has been suggested in this paper to diagnose bearing faults using imbalanced data samples and noisy data samples. In order to improve the diagnostic accuracy and anti-noise ability of the Capsnet model, multiple IMF signals that were obtained by the decomposition of a raw vibration signal that was based on CEEMDAN so as to capture more bearing fault information, were input into the Capsnet with CBAM in order to diagnose the fault. These preliminary diagnostic results were then fused through the weighted majority voting method, thus the final diagnostic decision was able to be obtained.

In order to validate the diagnostic effectiveness and anti-noise ability of the proposed ensemble Capsnet with CBAM, a multifaceted comparison with single CNN, single Capsnet, ensemble CNN and ensemble Capsnet without CBAM on five datasets with different imbalance degrees and on the same imbalanced dataset with different SNRs was implemented. The diagnostic results demonstrated that the proposed ensemble Capsnet with CBAM that is based on the weighted majority voting method can achieve outstanding diagnostic accuracy on these different imbalanced and SNR datasets and it is obviously superior to the other ensemble learning models for the diagnosis of bearing faults that are based on VM, single Capsnet without CBAM and single Capsnet with CBAM. Nevertheless, in the diagnostic process, it was found that the number of Capsnet models that were used in the ensemble affected the diagnostic performance, so the manner in which to select the number of Capsnet models for application to different IMF signals will be studied further in the future.

## Figures and Tables

**Figure 1 sensors-22-05543-f001:**
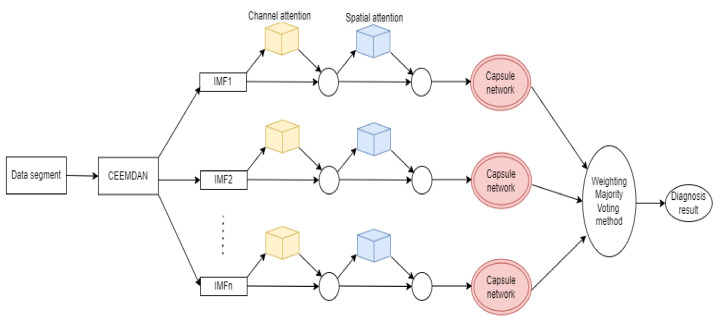
The diagnosis schematic of the ensemble capsule network.

**Figure 2 sensors-22-05543-f002:**
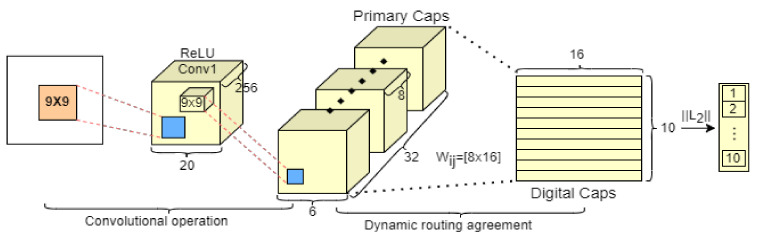
The basic architecture of Capsnet.

**Figure 3 sensors-22-05543-f003:**
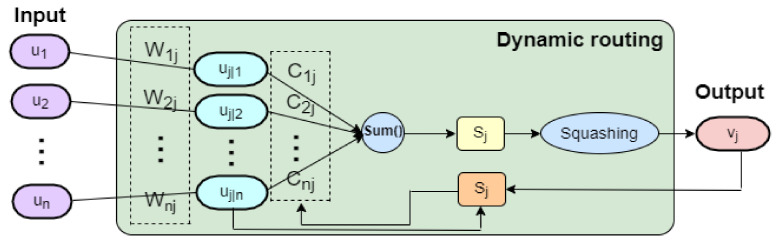
The dynamic routing algorithm in Capsnet.

**Figure 4 sensors-22-05543-f004:**
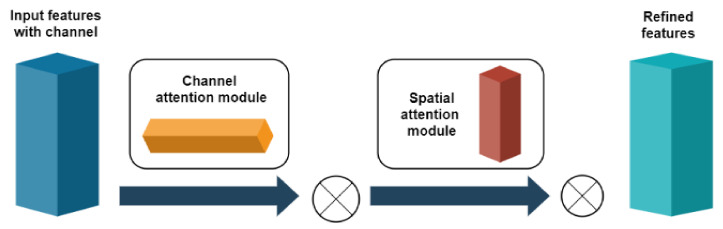
The structure of the CBAM attention mechanism.

**Figure 5 sensors-22-05543-f005:**
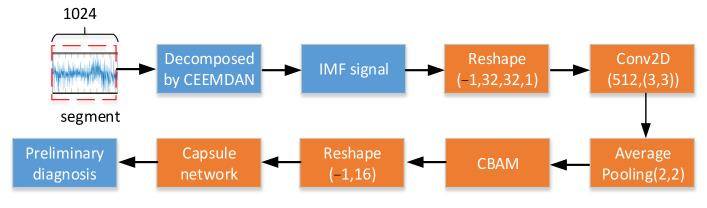
The architecture of Capsnet with CBAM.

**Figure 6 sensors-22-05543-f006:**
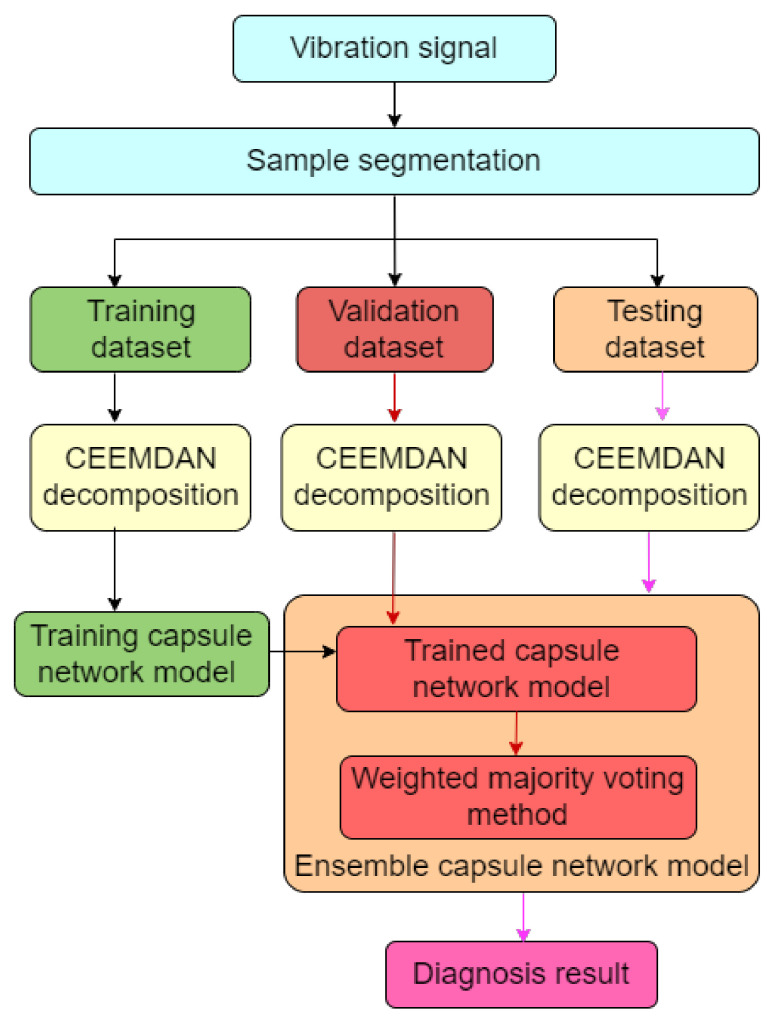
The diagnosis flowchart based on ensemble Capsnet.

**Figure 7 sensors-22-05543-f007:**
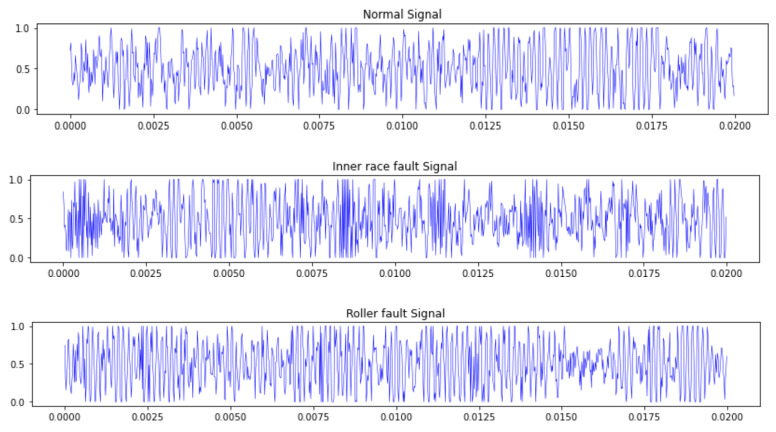
The original bearing vibration signals of different fault classes.

**Figure 8 sensors-22-05543-f008:**
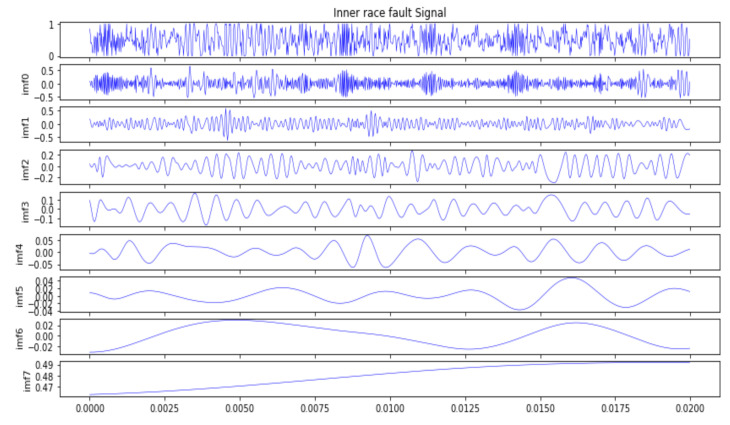
The decomposition results of inner race fault signal based on CEEMDAN.

**Figure 9 sensors-22-05543-f009:**
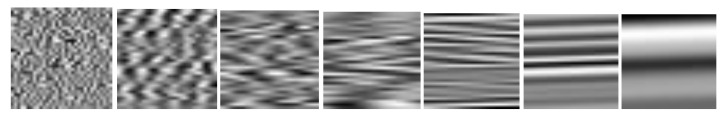
The corresponding grey images of the first seven orders of the IMF signal.

**Figure 10 sensors-22-05543-f010:**
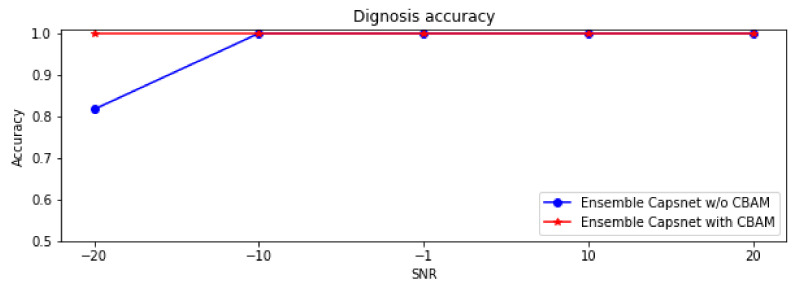
Diagnostic accuracy of ensemble Capsnet with CBAM/without CBAM on different SNR samples.

**Figure 11 sensors-22-05543-f011:**
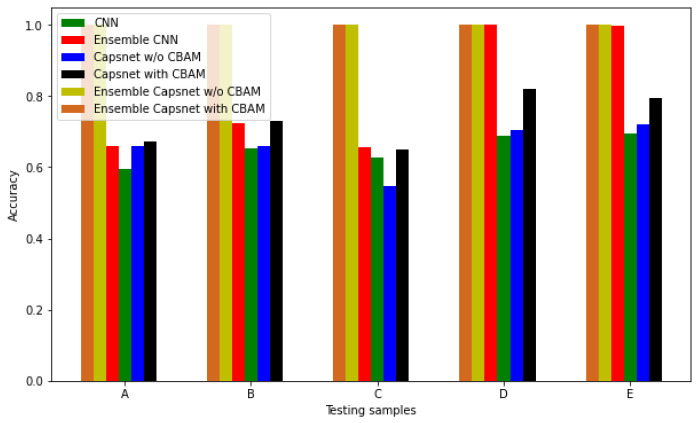
The diagnostic accuracy of six different diagnostic models.

**Figure 12 sensors-22-05543-f012:**
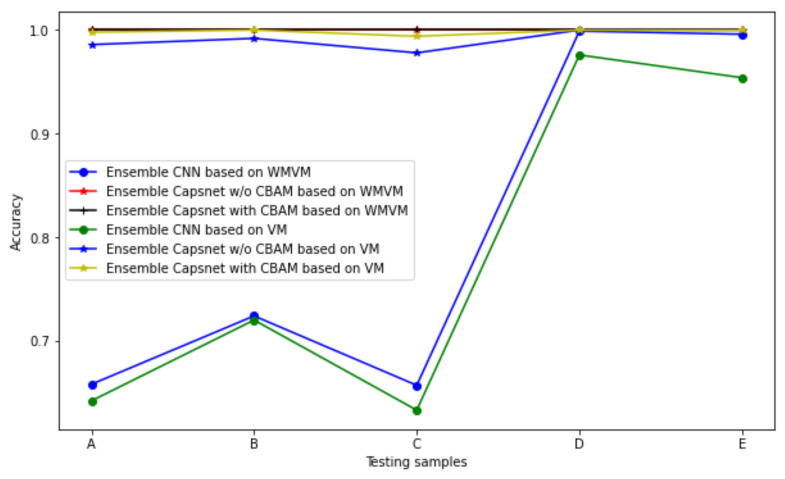
Diagnostic accuracy for different datasets based on different fusing methods.

**Figure 13 sensors-22-05543-f013:**
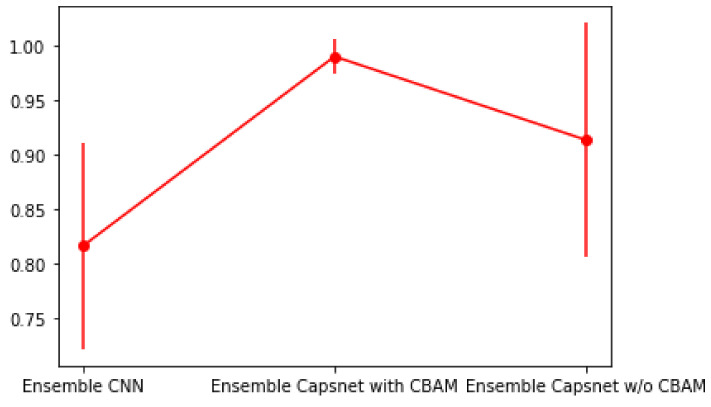
The diagnostic accuracy, mean and variance, of three different diagnostic models.

**Table 1 sensors-22-05543-t001:** The training/validation/testing data statistics of rolling bearings.

Dataset		Fault Category			Imbalance Ratio Normal:Fault
		Normal	Inner Race	Roller	
		Label:0 Speed:6000/12,000 Load:0/1012 N	Label:1 Speed:6000/12,000 Load:1006/1407 N	Label:2 Speed:6000/12,000 Load:992/1407 N	
Training/ validation samples	Dataset A	180/180	20/20	20/20	9:1
	Dataset B	280/280	20/20	20/20	14:1
	Dataset C	380/380	20/20	20/20	19:1
	Dataset D	200/200	40/40	40/40	5:1
	Dataset E	400/400	40/40	40/40	10:1
Testing samples	Dataset A, B, C, D and E	400	400	400	1:1

**Table 2 sensors-22-05543-t002:** The diagnostic accuracy of testing samples.

Dataset	Diagnostic Accuracy							
	IMF0	IMF1	IMF2	IMF3	IMF4	IMF5	IMF6	Ensemble Capsnet with CBAM
A	1	0.833	0.832	0.998	0.578	0.333	0.333	1
B	1	0.833	0.736	0.999	0.593	0.333	0.333	1
C	1	0.833	0.833	0.999	0.667	0.333	0.333	1
D	1	1	0.928	0.999	0.668	0.333	0.333	1
E	1	0.833	0.889	0.999	0.743	0.334	0.333	1

**Table 3 sensors-22-05543-t003:** The diagnostic accuracy of two diagnostic models on different SNR datasets.

		Testing Samples with Different SNR				
SNR(db)		−20	−10	−1	10	20
Diagnostic accuracy	Ensemble Capsnet with CBAM	1	1	1	1	1
	Ensemble Capsnet w/o CBAM	0.817	1	1	1	1

**Table 4 sensors-22-05543-t004:** The structural parameters of CNN.

No.	Layer	Activation Shape
1	Input layer	(None, 32, 32, 1)
2	Conv2D	(None, 32, 32, 128)
3	Average Pooling	(None, 16, 16, 128)
4	Conv2D	(None, 16, 16, 512)
5	Global average Pooling	(None, 512)
6	Dense	(None, 100)
7	Dense	(None, 3)

**Table 5 sensors-22-05543-t005:** The diagnostic accuracy of six diagnostic models for different datasets.

Testing Samples	Diagnostic Accuracy					
	CNN	Capsnet with CBAM	Capsnet w/o CBAM	Ensemble CNN	Ensemble Capsnet w/o CBAM	Ensemble Capsnet with CBAM
A	0.595	0.673	0.658	0.658	1	1
B	0.653	0.73	0.658	0.724	1	1
C	0.628	0.649	0.548	0.657	1	1
D	0.689	0.82	0.703	0.999	1	1
E	0.695	0.794	0.719	0.996	1	1

**Table 6 sensors-22-05543-t006:** The diagnostic accuracy of six different ensemble models for different datasets.

Testing Samples	Diagnostic Accuracy					
	Ensemble CNN Based on VM	Ensemble Capsnet with CBAM Based on VM	Ensemble Capsnet w/o CBAM Based on VM	Ensemble CNN Based on WMVM	Ensemble Capsnet with CBAM Based on WMVM	Ensemble Capsnet w/o CBAM Based on WMVM
A	0.642	0.998	0.986	0.658	1	1
B	0.720	0.998	0.992	0.724	1	1
C	0.633	0.994	0.978	0.657	1	1
D	0.976	1	1	0.999	1	1
E	0.954	1	1	0.996	1	1

## Data Availability

The data that were used to support this study are available at the Politecnico di Torino rolling bearing test rig [38].

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
