# Peer review of "Ensemble Capsule Network with an Attention Mechanism for the Fault Diagnosis of Bearings from Imbalanced Data Samples"

_sensors, 2022, doi:10.3390/s22155543_

Round 1

Reviewer 1 Report

This manuscript proposed an ensemble capsule network to bearing fault diagnosis when facing imbalanced data. The novelty of the proposed method relies on its convolutional block attention module (CBAM) and weighted majority voting method. Besides, the complete ensemble empirical mode decomposition with adaptive noise (CEEMDAN) method is introduced to decompose the vibration signals into features for further learning. Although the effectiveness of the proposed method was verified by experimental analysis, some problems still exist in this manuscript. Comments are as follows.

1.      The proposed method includes the feature extraction module, the feature learning module and the weighted voting module. The feature extraction is realized by the CEEMDAN. The feature learning is realized by the CBAM. However, weighted voting is a commonly used strategy to deal with imbalanced classification problems. The proposed method seems to just simply combine the existing technologies.

2. The method aims to diagnose faults under imbalanced fault data. However, to verify the effectiveness of the proposed method, the authors should compare the proposed method with existing imbalanced diagnosis models and write down the main difference and advantage.

3. It is better to use natural faults in the physical experiment. Artificial faults normally produce clean fault signals that can be easily handled. Besides, the authors do not show temporal and spectra signals so that it is hard to judge the difficulty of the provided case study.

 In general, the paper is well written but require modification suggest from above.

Author Response

  1. The proposed method includes the feature extraction module, the feature learning module and the weighted voting module. The feature extraction is realized by the CEEMDAN. The feature learning is realized by the CBAM. However, weighted voting is a commonly used strategy to deal with imbalanced classification problems. The proposed method seems to just simply combine the existing technologies.

A The proposed method is mainly to use Capsnet to extract features from different IMF signal decomposed by CEEMDAN, which can obtain some different-scale features concealed in the original vibration. Furthermore, different IMF signals have different contribution to the diagnosis results, weighted voting method can give different weights to different IMF signal according to the different contribution. So the proposed method is not simply combined technologies.

  1. The method aims to diagnose faults under imbalanced fault data. However, to verify the effectiveness of the proposed method, the authors should compare the proposed method with existing imbalanced diagnosis models and write down the main difference and advantage.

A: The proposed method is different from existing imbalanced diagnosis models such as GAN and over-sampling which are called data augmentation methods, the proposed method does not generate additional new data samples, so, the proposed method is only compared with ensemble CNN and ensemble Capsnet based on voting methods.   

  1. It is better to use natural faults in the physical experiment. Artificial faults normally produce clean fault signals that can be easily handled. Besides, the authors do not show temporal and spectra signals so that it is hard to judge the difficulty of the provided case study.

A: Considering these bearing fault signals are representative and widely used to validate many diagnosis models. So it is utilized to verify the effectiveness of the proposed method. In addition, the temporal signals of different bearing fault classes are shown in the paper.

Reviewer 2 Report

An ensemble capsule network with a convolutional block attention module based on weighted majority voting method is designed in this paper. The experimental verification is clear and the comparisons are complete. However, there are still some issues:

1.     The contributions 2) and 3) are just the description of the effectiveness of your proposed method, and there are in fact not your main contributions. Thus, the authors should rewrite this part to make your contributions clearer.

2.     The authors should list the hyper-parameters of the network parameters and the training details, which is very. Important for the reproduction.

3.      You can compare your method with other state-of-the-art models, please refer to deep learning algorithms for rotating machinery intelligent diagnosis.

4.     The format of the references should be unified.

5.     Some abbreviations, which firstly appeared in this paper, such as CNN, ANN, etc. should be written in the full name.

Author Response

  1. The contributions 2) and 3) are just the description of the effectiveness of your proposed method, and there are in fact not your main contributions. Thus, the authors should rewrite this part to make your contributions clearer.

A: The contribution are revised accordingly .  

  1. The authors should list the hyper-parameters of the network parameters and the training details, which is very Important for the reproduction.

A: The structure parameters of single Capsnet with CBAM is shown in Fig.5, and the structure parameters of CNN is added to the revised paper, which is shown in Table 4.

  1. You can compare your method with other state-of-the-art models, please refer to deep learning algorithms for rotating machinery intelligent diagnosis.

A: Owing to the motivation is to use ensemble Capsnet to diagnose the imbalanced data samples when the new data samples are not generated, the ensemble Capsnet is only compared with other ensemble CNN, ensemble Capsnet without CBAM, single CNN and single Capsnet, and it is not compared with other deep learning such as GAN and its invariants.

  1. The format of the references should be unified.

A: The reference has been formatted.  

  1. Some abbreviations, which firstly appeared in this paper, such as CNN, ANN, etc. should be written in the full name.

A: Full names of abbreviation have been provided in Section 1.

Reviewer 3 Report

Authors proposed methodology based on Ensemble capsule network for bearing fault diagnosis under strong noise. Fault diagnosis results are encouraging to the readers, however there are certain issues which need justifications in revised manuscript :  

1. Title should be modify. For example : Ensemble capsule network with attention (mechanism/algorithm etc). Similarly, classification of bearing fault diagnosis under strong noise . Do there is a need to use word " Noise "  in title ?

2. Affiliation detail of authors are incomplete.

3. Authors mentioned in pg.2 line 54-55 " some drawbacks for these GAN models, such as gradient vanishing, mode collapse, etc., which can reduce the quality of the generated samples ". Do these drawbacks are addressed with the proposed methodology ?

4. It is not justified to use all extracted IMFs from EEMD/VMD etc. for fault diagnosis purpose.Selection of suitable IMF is crucial for unbiased fault diagnosis accuracy.Authors should refer following journals and include the discussion in revised manuscript :

a. https://journals.sagepub.com/doi/abs/10.1177/09544062211043132.

b. https://www.mdpi.com/1099-4300/24/7/927.

c. https://link.springer.com/chapter/10.1007/978-3-030-00184-1_10

5. Resolution of figures should be improved.

6. The contribution of authors mentioned in pg.3 from line 109-117 should be more clear.

7. It is suggested to include some recent literatures related to GAN,Capsule network etc. in revised manuscript with sufficient discussions.

8. Variations in formatting styles of references are observed.For ex: 19,20 etc. Kindly rectify the variations.

Author Response

  1. Title should be modify. For example : Ensemble capsule network with attention (mechanism/algorithm etc). Similarly, classification of bearing fault diagnosis under strong noise . Do there is a need to use word " Noise "  in title ?

A: The title has been revised as “Ensemble capsule network with attention mechanism for the fault diagnosis of bearing with imbalanced data samples”.

  1. Affiliation detail of authors are incomplete.

A: They are revised in the paper.

  1. Authors mentioned in pg.2 line 54-55 " some drawbacks for these GAN models, such as gradient vanishing, mode collapse, etc., which can reduce the quality of the generated samples ". Do these drawbacks are addressed with the proposed methodology ?

A: The proposed methodology cannot solve the drawbacks of the GAN models.  But this part in the introduction is revised.

  1. It is not justified to use all extracted IMFs from EEMD/VMD etc. for fault diagnosis purpose. Selection of suitable IMF is crucial for unbiased fault diagnosis accuracy. Authors should refer following journals and include the discussion in revised manuscript:
  2. https://journals.sagepub.com/doi/abs/10.1177/09544062211043132.
  3. https://www.mdpi.com/1099-4300/24/7/927.
  4. https://link.springer.com/chapter/10.1007/978-3-030-00184-1_10

A: we referred these three papers in the revised paper. Selection of suitable IMF is crucial for unbiased fault diagnosis accuracy indeed. So, after that all IMF signals are fed into Capsnet to diagnose the fault, the weighted voting method is utilized to fuse all the preliminary diagnosis results according to the weight coefficients which are obtained by the importance degree of IMF to the diagnosis results.

  1. Resolution of figures should be improved.

A: Resolution of figures are improved in the manuscript.

  1. The contribution of authors mentioned in pg.3 from line 109-117 should be clearer.

A: They are revised in the manuscript.

  1. It is suggested to include some recent literatures related to GAN, Capsule network etc. in revised manuscript with sufficient discussions.

A: They are revised in the manuscript. Considering the motivation of the paper, GAN is only used to demonstrate that data augmentation methods based on GAN have some general drawback, so it is not discussed briefly in the revised paper. But the Capsule network is discussed in detail in the revised manuscript.

  1. Variations in formatting styles of references are observed. For ex: 19,20 etc. Kindly rectify the variations.

A: They are all revised in the manuscript.

Reviewer 4 Report

This paper resents a capsule network with attention for bearing fault diagnosis under unbalanced data and strong noise. To perform the method, the complete ensemble EMD is used for signal preprocessing. The paper provides related technical details and experimental validation. However, some major issues remain unsolved.

In this research, it appears that many components/modules are integrated into the framework. The framework looks like very “fancy” and “complex” in terms of its architecture. However, it is not always true that more complexity makes better performance. At this point, authors are required to explain the questions like: why did authors decide to use capsule network, CEMDAN and so on? any advantages of those components? Why did authors feel confident that putting them together can boost the performance?  These need to be clarified from a fundamental perspective. I recommend revising the Introduction thoroughly to highlight the answers.

Addressing the unbalanced data issue in fault diagnosis is not new. Other challenging fault diagnosis mission, such as extended fault diagnosis under limited labels, fault diagnosis under very severe time-varying conditions and other uncertainties are also practically important. The authors need to discuss relevant applications/works in the first paragraph of Introduction. Some examples are given as

https://doi.org/10.1109/TIE.2015.2460242

https://doi.org/10.1016/j.jsv.2017.03.037

https://doi.org/10.1109/TIE.2009.2016517

https://doi.org/10.1007/s00170-021-07253-6

I am curious why the authors used CEMDAN instead of EMD. EMD also is effective in handling the nonstationary signal with system nonlinearity and reducing the noise where possible. Again, pursuing the more sophisticated algorithm is not always a good choice. This needs to be clarified.

Caption of Figure 6 is way off the figure.

The different features in Section 2.2 reads like independent to the rest. There correlations/connections need to be clearly mentioned.

In section 2.2.1, authors described the convolutional operation in detail but with very few narratives to describe the dynamic routing agreement. More details regarding the dynamic routing agreement should be provided as this is the key attribute to distinguish the regular CNN and Capsnet.

Figure 8 is hard to read. The label font is very small. The figure quality is bad. The unit of horizontal axis is not given. It should be revised for quality improvement. Following the above comment, if authors used EMD, what will the result (Figure 8) look like. Based on my experience, IMFs obtained from EMD should have very similar patterns.

Authors mentioned “the 7 different single 370 Capsnet diagnosis models, whose hyper-parameters are different, can be obtained.” What is the strategy to tune the hyperparameters? The relevant details are missing.  Hyperparameter tuning always is the key to neural network performance.

The takeaway of Table 2 result is unclear. Using the first IMF, the accuracy already is 100% (because the first IMF is most discriminative to fault), so what is the purpose to further compare the Capsnet ensemble? In other words, is such comparison meaningful?

In Section 3.4., many models are not explained. For example, CNN is involved. What is particular structure of CNN? How many layers are involved? Try to think about how to make rigorous and fair comparison as there are so many different hyperparameters in different models.

When authors presented the result comparison, only one set of results for each model and each dataset is given. Considering the randomness in model training and data split, cross validation is needed to comprehensively assess the model performance robustness.

Resolution of Figure 10 is quite low, which needs to be improved.

Almost all result figures have very small fonts, which need to be enlarged.

Paper should undergo serious proofreading to improve the language and avoid grammar mistakes.

Author Response

 1.In this research, it appears that many components/modules are integrated into the framework. The framework looks like very “fancy” and “complex” in terms of its architecture. However, it is not always true that more complexity makes better performance. At this point, authors are required to explain the questions like: why did authors decide to use capsule network, CEMDAN and so on? any advantages of those components? Why did authors feel confident that putting them together can boost the performance?  These need to be clarified from a fundamental perspective. I recommend revising the Introduction thoroughly to highlight the answers.

A: Ensemble learning can overcome the limitations of data samples imbalance and achieves high diagnosis accuracy and generalization because of the complementary diagnosis behaviour among different diagnosis models without generating additional fault data samples, CEMDAN can decompose the vibration signal into different IMF components and reduce the mode mixing of IMF components. Capsule network can not only extract the features from the IMF signal but also conserve the spatio-temporal relationship of different features. CBAM can select some sensitive features. So, when all these modules are integrated into the ensemble Capsnet with CBAM, all advantages of these modules can boost the diagnosis performance.

Furthermore, the introduction has been revised according to this suggestion. 

 2.Addressing the unbalanced data issue in fault diagnosis is not new. Other challenging fault diagnosis mission, such as extended fault diagnosis under limited labels, fault diagnosis under very severe time-varying conditions and other uncertainties are also practically important. The authors need to discuss relevant applications/works in the first paragraph of Introduction. Some examples are given as

https://doi.org/10.1109/TIE.2015.2460242

https://doi.org/10.1016/j.jsv.2017.03.037

https://doi.org/10.1109/TIE.2009.2016517

https://doi.org/10.1007/s00170-021-07253-6

A: The relevant discussion is added to the first paragraph of Introduction, and the references are added at the end of the revised paper.

  1. I am curious why the authors used CEMDAN instead of EMD. EMD also is effective in handling the nonstationary signal with system nonlinearity and reducing the noise where possible. Again, pursuing the more sophisticated algorithm is not always a good choice. This needs to be clarified.

A: This is revised in the introduction. It is mainly because that the CEEMDAN can reduce the mode mixing of different IMF components.

4.Caption of Figure 6 is way off the figure.

 A: It is revised in the manuscript.

  1. The different features in Section 2.2 reads like independent to the rest. There correlations/connections need to be clearly mentioned.

A: Section 2.2 is revised in the paper and highlighted in red colour.

  1. In section 2.2.1, authors described the convolutional operation in detail but with very few narratives to describe the dynamic routing agreement. More details regarding the dynamic routing agreement should be provided as this is the key attribute to distinguish the regular CNN and Capsnet.

 A: Thus, with the iterations from (6) to (12), all these parameters and the agreement-based dynamic routing algorithm of Capsnet are determined.

  1. Figure 8 is hard to read. The label font is very small. The figure quality is bad. The unit of horizontal axis is not given. It should be revised for quality improvement. Following the above comment, if authors used EMD, what will the result (Figure 8) look like. Based on my experience, IMFs obtained from EMD should have very similar patterns.

A: Figure 8 is revised accordingly.

  1. Authors mentioned “the 7 different single 370 Capsnet diagnosis models, whose hyper-parameters are different, can be obtained.” What is the strategy to tune the hyperparameters? The relevant details are missing.  Hyperparameter tuning always is the key to neural network performance.

A: The structure and layer number of the single Capsnet are all the same, but each Capsnet model is trained by different order IMF signal respectively. So, the corresponding hyper-parameters of each trained Capsnet are different.

  1. The takeaway of Table 2 result is unclear. Using the first IMF, the accuracy already is 100% (because the first IMF is most discriminative to fault), so what is the purpose to further compare the Capsnet ensemble? In other words, is such comparison meaningful?

A: The main purpose is to verify the diagnosis performance of ensemble Capsnet such as high accuracy and strong generalization.

  1. In Section 3.4., many models are not explained. For example, CNN is involved. What is particular structure of CNN? How many layers are involved? Try to think about how to make rigorous and fair comparison as there are so many different hyperparameters in different models.

 A: The structure parameters of CNN are added to the revised paper.

  1. When authors presented the result comparison, only one set of results for each model and each dataset is given. Considering the randomness in model training and data split, cross validation is needed to comprehensively assess the model performance robustness.

A: Cross validation has been done in the revised paper.

12.Resolution of Figure 10 is quite low, which needs to be improved.

 A: It is revised in the manuscript.

  1. Almost all result figures have very small fonts, which need to be enlarged.

 A: Figures with small fonts have been adjusted.

  1. Paper should undergo serious proofreading to improve the language and avoid grammar mistakes.

A: The paper has been proof-read.

Round 2

Reviewer 2 Report

The author has well answered my comments.

Reviewer 3 Report

Authors addressed all reviewer comments with proper justifications and accordingly modified the manuscript.

Reviewer 4 Report

Authors addressed my comments.